# Classification of symptom subtypes in patients with multiple myeloma during treatment: a cross-sectional survey study in China

Chunfang Yu,[1] Tingting Cai,[2] Tingting Zhou,[2] Ning Zeng,[3] Xin Liang,[4] Guihua Pan,[3] Wei Ouyang,[5] Changrong Yuan [iD] [2]

CYu and TC contributed equally.

CYu and TC are joint first authors.

For numbered affiliations see end of article.

**Correspondence to**
Dr Changrong Yuan;
yuancr@fudan.edu.cn

## ABSTRACT

**Objectives** To classify subgroups of cancer-related symptoms in patients with multiple myeloma (MM) during treatment and examine between-group differences in demographic and clinical characteristics in addition to functional status.

**Design** Cross-sectional survey study.

**Setting** Haematology department of two tertiary hospitals affiliated with Guilin Medical University in China.

**Participants** Using a convenience sampling method, questionnaires were distributed to patients with MM visiting two hospitals in Guilin, China.

**Interventions** The patients were categorised into subgroups based on cancer-related symptoms using a latent class analysis. An analysis of covariance was performed to examine how demographic and clinical characteristics and functional status differed among the subgroups.

**Results** In total, 216 patients completed the survey, with an average age of 60.3 years. A three-class solution was identified: low symptom burden group (class 1, 36.6%), moderate symptom burden group (class 2, 34.2%) and high symptom burden group (class 3, 29.2%). Patients with low monthly family income (OR=3.14, p=0.010) and complications of MM bone disease (OR=2.95, p=0.029) were more likely to belong to class 2. The predictors of high-burden symptoms were treated with painkillers, antidepressants or hypnotic drugs (OR=3.68, p=0.012) and <5000 daily step counts (OR=2.52, p=0.039) in class 3. Functional status was correlated with symptom burden, with patients in classes 3 and 1 reporting significantly higher and lower functional status, respectively (p<0.05).

**Conclusions** Patients with MM experienced varying degrees of symptoms during treatment. The identification of patients with high symptom burden management should focus on the assessment of demographic and clinical characteristics, in addition to functional status.

## STRENGTHS AND LIMITATIONS OF THIS STUDY

⇒ A latent class analysis was used to categorise cancer-related symptoms in Chinese patients undergoing treatment for multiple myeloma (MM).

⇒ This study assessed between-group differences in the demographic and clinical characteristics and functional status of patients with MM.

⇒ The Patient-Reported Outcomes Measurement Information System (PROMIS)-57 and the PROMIS Cognitive Function Short Form were used as research tools.

⇒ This study involved a limited number of treatment centres.

⇒ Patient clinical characteristics did not include the evaluation of treatment response.

age with a median age at diagnosis of 73 years.[1] Older patients have a higher rate of comorbid chronic and progressive diseases than younger patients, and, compared with younger patients, a more careful approach is required to older patients during MM treatment to manage multiple treatment-related issues, including symptoms and functional limitations.[2] Patients with MM frequently report symptoms, including pain, fatigue and depression, after diagnosis and throughout treatment.[3] With increasing survival rates, increased attention has focused on symptom control in this population.

Although advancements in MM treatment have been extensively reported, little is known about the symptoms of patients with MM during treatment.[4–6] Monitoring symptoms complements traditional laboratory testing and contributes to the identification of individuals who require supportive interventions. The effects of MM and its treatment may have an effect on patients' physical and emotional well-being, in addition to their social function. There is a paucity of data regarding the symptoms experienced by patients with MM,

## INTRODUCTION

Multiple myeloma (MM) is the second most common haematological cancer worldwide.[1] Although still incurable, advancements in the management of MM have extended survival, and MM is now considered a chronic disease. MM is primarily a cancer of old

particularly those undergoing therapy. Symptoms may affect cancer treatment options and outcomes. Understanding individual symptom experiences is important to identify patients who are at risk of manifesting severe symptoms.

Evidence suggests that direct patient reports are the best way to capture information on patient experience of disease and related treatment.[2 6 7] Patient-reported outcomes (PROs) are strongly recommended for the measurement of symptoms and treatment effectiveness in patients with MM.[7] Specifically, among patients undergoing therapy, PROs can provide a better understanding of the magnitude and temporal trends of symptoms from the perspective of individual patients. Additionally, using PROs to capture patient symptoms is necessary to perform value-based and patient-centred care and to enable more accurate symptom evaluation.[7]

To date, the heterogeneity of symptoms has not been captured among patients with MM. Patients experience multiple symptoms that potentially interact with each other and that occur in clusters of clinically distinct subgroups. Statistical advancements in modelling contribute to the identification of subtypes and may be useful in distinguishing patients at risk. Latent class analysis (LCA) is a person-oriented approach used to classify individuals based on their symptom profiles and estimate distinct patterns.[8]

There is no existing study to explore symptom profiles resulting from MM and its treatment in this population. Therefore, this study aimed to classify subgroups of patients with MM by incorporating PRO instruments and to examine between-group differences in demographic and clinical characteristics, in addition to functional status.

## METHODS
### Study design and sample
This study adopted a cross-sectional study design and adhered to the Strengthening the Reporting of Observational Studies in Epidemiology guidelines. Convenience sampling was used to recruit patients with MM visiting the haematology department of two tertiary hospitals affiliated with Guilin Medical University in China between July and December 2021. Eligible patients were aged >18 years, diagnosed with primary MM, hospitalised for treatment, and able to speak and write Mandarin Chinese. Patients with other types of cancer or mental illnesses were excluded from this study.

### Study instruments
#### Demographic and clinical data
Demographic and socioeconomic data included age, sex, marital status, educational background, employment status, monthly family income, medical insurance, daily sleep duration and daily step counts. The MM type, disease stage, drug usage, complications, treatments and body mass index (BMI) were retrieved from the medical records.

### Patient-Reported Outcomes Measurement Information System-57
Symptom experiences were measured using the Patient-Reported Outcomes Measurement Information System (PROMIS)-57 instrument. The PROMIS-57 consists of 57 items and includes five symptoms (anxiety, depression, fatigue, sleep disturbance, pain interference) and two functional domains (physical function, ability to participate in social roles and activities), along with one additional pain intensity item.[9] The instrument adopts a Likert scale (0–5) to assess the severity of symptoms or functional limitation, except for pain intensity, which has 11 response options (ranging from 0 to 10). The raw score of all outcome scores ranged from 4 to 20 and was converted into T-scores with a mean of 50 and an SD of 10 in the scoring manual.[10] A higher score indicates poor functional status; however, for the symptom domain, this is reversed, with higher scores reflecting a high symptom burden.[9] Cronbach's α coefficient ranged from 0.85 to 0.95 in this study.

### PROMIS-Cognitive Function Short Form
The four-item PROMIS-Cognitive Function Short Form measures assessed cognitive function over the preceding 7 days. Each question was given a response of 1 ('never') to 5 ('very often').[11] The total raw scores ranged from 4 to 20 and were converted into T-scores. Similar to the PROMIS-57, lower T-scores indicate greater subjective cognitive difficulty. The Cronbach's α for the scale was 0.93.

### Data collection
Eligible patients were approached and provided with information about the study by assistant nurses. Both online and offline surveys were used to collect the data, and the assistant nurses assessed the questionnaires for completeness.

### Statistical analyses
Statistical analyses were performed using the SPSS V.22.0 and Mplus V.8.0. Categorical and continuous variables are expressed as frequencies and proportions and medians with IQRs, respectively. The models were run from one to six classes to determine the optimal number of latent classes. The model fit for LCA included the Akaike information criterion (AIC), Bayesian information criterion (BIC), sample size adjusted BIC (aBIC), entropy, and the Lo-Mendell-Rubin likelihood-ratio test (LMR-LRT). Lower AIC, BIC and aBIC, higher entropy, and significant LMR-LRT indicate a better model fit.[12 13]

After identifying the optimal latent class model, the symptom probabilities of identified subgroups were depicted by a line chart to present the characteristics of each group. To further identify between-group difference among the subgroups, an analysis of covariance was performed to examine how demographic, clinical characteristics and functional status differed among the

**Table 1** Demographic and clinical characteristics of the sample (n=216)

| Variable | n (%) |
|---|---|
| **Sex** | |
| Male | 87 (40.3) |
| Female | 129 (59.7) |
| **Marital status** | |
| Married | 199 (92.1) |
| Separated/divorced/widowed | 17 (7.9) |
| **Educational background** | |
| Junior high school or below | 153 (70.8) |
| High school | 35 (16.2) |
| University or above | 28 (13.0) |
| **Employment status** | |
| Employed | 14 (6.5) |
| Medical leave | 10 (4.6) |
| Unemployed | 68 (31.5) |
| Retired | 82 (38.0) |
| Others | 42 (19.4) |
| **Monthly family income (¥)** | |
| <5000 | 166 (76.9) |
| ≥5000 | 50 (23.1) |
| **MM type** | |
| IgG type | 119 (55.1) |
| IgA type | 79 (36.6) |
| IgD type | 9 (4.2) |
| Light chain type | 7 (3.2) |
| Others | 2 (0.9) |
| **Disease staging** | |
| New diagnosis | 13 (6.0) |
| Induction chemotherapy | 91 (42.1) |
| Consolidation chemotherapy | 62 (28.7) |
| Platform stage | 27 (12.5) |
| Unclear | 7 (3.2) |
| Recurrence | 16 (7.4) |
| **Use of painkillers, antidepressants or hypnotic drugs** | |
| No | 169 (78.2) |
| Yes | 47 (21.8) |
| **Number of complications** | |
| 0 | 5 (2.3) |
| 1 | 145 (67.1%) |
| 2 | 44 (20.4%) |
| ≥3 | 22 (10.2%) |
| **Types of complications** | |
| Multiple myeloma bone disease | 97 (44.9) |
| Impairment of renal function | 32 (14.8) |
| Anaemia | 71 (32.9) |

Continued

**Table 1** Continued

| Variable | n (%) |
|---|---|
| Hypercalcaemia | 5 (2.3) |
| Thrombosis | 5 (2.3) |
| Inflammation | 40 (18.5) |
| Other complications | 53 (24.5) |
| **Treatment** | |
| Oral drug therapy | 8 (3.7) |
| Chemotherapy | 170 (78.7) |
| Autologous haematopoietic stem cell transplantation | 19 (8.8) |
| Others | 19 (8.8) |
| **BMI** | |
| <18.5 | 14 (6.5) |
| 18.5–23.9 | 136 (63.0) |
| 24–27.9 | 52 (24.0) |
| >27.9 | 14 (6.5) |
| **Daily step counts** | |
| <5000 | 154 (71.3) |
| ≥5000 | 62 (28.7) |

BMI, body mass index; MM, multiple myeloma.

subgroups using the $\chi^2$ and Fisher's exact tests. Subsequently, a multivariate logistic regression model was used to identify the factors associated with different classes, and the results were reported as adjusted ORs with 95% CIs. Additionally, a comparison of functional status was performed using one-way analysis of variance (ANOVA). Statistical significance was defined as $p<0.05$.

### Patient and public involvement
During the design and preparation of the data collection tool, no study participants were involved in the development of the research question or the outcome measures.

## RESULTS
### Patient characteristics
The demographic and clinical characteristics of the patients are reported in table 1. In total, 216 patients were available for the analysis, with an average age of 60.3 (SD=10.4 (range=22–84)) years. Most of the patients were female (59.7%), married (92.1%) and retired (38.0%), with an educational level of junior high school or below (70.8%) and a monthly family income below ¥5000 (76.9%). In terms of clinical characteristics, most patients had IgG MM (55.1%), were undergoing chemotherapy (70.8%) and had multiple concurrent complications of treatment (97.7%). Regarding treatment options, most patients underwent chemotherapy (78.7%) and did not require analgesia, antidepressants or hypnotics before the diagnosis (97.7%). Most patients had a BMI within

**Table 2** Model fit statistics for each of the fitted latent class analysis models

| Model | AIC | BIC | aBIC | Entropy | LMR p value | BLRT p value | Classification probability |
|---|---|---|---|---|---|---|---|
| 2C | 1183.08 | 1220.21 | 1185.35 | 0.800 | <0.001 | <0.001 | 0.505/0.495 |
| 3C | 1169.58 | 1226.96 | 1173.08 | 0.710 | <0.001 | <0.001 | 0.366/0.342/0.292 |
| 4C | 1178.47 | 1256.10 | 1183.22 | 0.811 | 0.491 | 0.667 | 0.199/0.287/0.148/0.366 |
| 5C | 1186.50 | 1284.38 | 1192.49 | 0.786 | 0.582 | 1.000 | 0.292/0.278/0.314/0.019/0.347 |
| 6C | 1197.35 | 1315.49 | 1204.58 | 0.770 | 0.332 | 1.000 | 0.037/0.032/0.356/0.269/0.292/0.014 |

aBIC, adjusted Bayesian information criterion; AIC, Akaike information criterion; BIC, Bayesian information criterion; BLRT, bootstrap likelihood ratio test; LMR, Lo-Mendell-Rubin.

the normal range (63.0%) and reported daily step counts <5000 steps (71.3%).

### Latent class analysis

The detailed results of the model fitting statistics for one to six classes are presented in table 2. Among the types of model solutions, the three-class solution had the best fit (lowest AIC and BIC values), high entropy, significant p values for LMR and BLRT, and optimal clinical interpretability of each latent class. Therefore, the three-class solution was selected as the optimal solution.

### Classification of symptoms and functional status

The item probabilities of the latent classes are shown in online supplemental figure 1. Class 1 was characterised by low item probabilities for all symptom aspects and was labelled as the 'low symptom burden group' (n=79, 36.6%). Class 2 was characterised by elevated item probabilities for anxiety, depression and pain interference, whereas decreased item probabilities were associated with fatigue and sleep disturbance. Therefore, class 2 was labelled as the 'moderate symptom burden group' (n=74, 34.2%). Class 3 was characterised by high item probabilities for all symptoms and was labelled as the 'high symptom burden group' (n=63, 29.2%).

### Between-group differences in symptom status across the latent classes

A comparison of symptoms across the latent classes is reported in table 3. The T-scores for anxiety, depression, fatigue, sleep disturbance and pain interference were relatively lower in the sample according to the score manual of PROMIS-57 compared with the general population. There were significant differences in the mean T-scores for each domain among the latent classes, with

the highest scores observed in class 3 and the lowest scores observed in class 1 (p<0.05).

### Demographic and clinical characteristics across the latent classes

Demographic and clinical characteristics of the latent classes are described in table 4. The results of the $\chi^2$ tests indicated that the proportion of patients assigned to the identified latent classes varied in terms of monthly family income (p=0.001); use of painkillers, antidepressants or hypnotic drugs (p<0.001); MM bone disease (p<0.001); anaemia (p=0.014); other complications (p=0.017) and daily step counts (p=0.043). ANOVA results indicated that there were no significant differences in daily sleep time (p=0.014) among the identified latent classes. No significant differences were identified for other variables.

Based on the outcomes of univariate analysis, a multinomial logistic regression analysis demonstrated that patients with low monthly family income (OR=3.14, p=0.010) and complications of MM bone disease (OR=2.95, p=0.029) were more likely to belong to class 2. Additionally, patients treated with analgesics, antidepressants or hypnotic drugs (OR=3.68, p=0.012) and those with daily step counts <5000 steps (OR=2.52, p=0.039) had a higher predisposition toward class 3, which was also characterised by a high incidence of symptoms. Adequate daily sleep time (OR=0.74, p=0.031) played a protective role in avoiding a high incidence of symptoms (table 5).

### Between-group differences in functional status across the latent classes

A comparison of functional status across the latent classes is reported in table 6 and online supplemental figure 2. The T-scores for physical function, ability to participate in social roles and activities, and cognitive function were relatively lower in the sample according to the score manual of PROMIS-57, in which physical function was slightly impaired, compared with the general population. Significant differences were identified among the three classes, with the highest scores in all functional dimensions observed in class 1 and the lowest scores observed in class 3 (p<0.05).

**Table 3** Symptoms across the latent classes (mean±SD)

| Domain | Class 1 (n=79) | Class 2 (n=74) | Class 3 (n=63) | P value |
|---|---|---|---|---|
| Anxiety | 44.0±8.0 | 56.9±4.8 | 64.3±6.7 | <0.001 |
| Depression | 43.9±7.3 | 56.5±4.5 | 62.7±6.4 | <0.001 |
| Fatigue | 43.1±7.2 | 51.5±3.7 | 61.2±5.6 | <0.001 |
| Sleep disturbance | 50.6±6.8 | 51.0±3.8 | 55.4±3.7 | <0.001 |
| Pain interference | 48.0±9.5 | 57.5±5.5 | 64.7±5.2 | <0.001 |

**Table 4** Comparisons of demographic and clinical characteristics among the latent classes

| Variable | Class 1 (n=79) | Class 2 (n=74) | Class 3 (n=63) | P value |
|---|---|---|---|---|
| Age (mean±SD), years | 58.7±10.4 | 61.5±9.5 | 60.8±11.4 | 0.239 |
| Sex | | | | 0.258 |
| Male | 35 (44.3%) | 32 (43.2%) | 20 (31.7%) | |
| Female | 44 (55.7%) | 42 (56.8%) | 43 (68.3%) | |
| Marital status | | | | 0.320 |
| Married | 8 (10.1%) | 3 (4.1%) | 6 (9.5%) | |
| Separated/divorced/widowed | 71 (89.9%) | 71 (95.9%) | 57 (90.5%) | |
| Educational background | | | | 0.477 |
| Junior high school or below | 53 (67.1%) | 54 (73.0%) | 46 (73.0%) | |
| High school | 12 (15.2%) | 14 (18.9%) | 9 (14.3%) | |
| University or above | 14 (17.7%) | 6 (8.1%) | 8 (12.7%) | |
| Employment status | | | | 0.814 |
| Employed | 6 (7.6%) | 5 (6.8%) | 3 (4.8%) | |
| Medical leave | 2 (2.5%) | 5 (6.8%) | 3 (4.8%) | |
| Unemployed | 25 (31.7%) | 23 (31.1%) | 20 (31.7%) | |
| Retired | 28 (35.4%) | 31 (41.8%) | 23 (36.5%) | |
| Others | 18 (22.8%) | 10 (13.5%) | 14 (22.2%) | |
| Monthly family income (¥) | | | | 0.001 |
| <5000 | 50 (63.3%) | 64 (86.5%) | 52 (82.5%) | |
| ≥5000 | 29 (36.7%) | 10 (13.5%) | 11 (17.5%) | |
| MM type | | | | 0.797 |
| IgG type | 42 (53.1%) | 43 (58.1%) | 34 (54.0%) | |
| IgA type | 28 (35.4%) | 27 (36.4%) | 24 (38.1%) | |
| IgD type | 4 (5.1%) | 1 (1.4%) | 4 (6.3%) | |
| Light chain type | 4 (5.1%) | 2 (2.7%) | 1 (1.6%) | |
| Others | 1 (1.3%) | 1 (1.4%) | 0 (0.0%) | |
| Disease stage | | | | 0.056 |
| New diagnosis | 6 (7.6%) | 2 (2.7%) | 5 (7.9%) | |
| Induction chemotherapy | 34 (43.1%) | 31 (41.9%) | 26 (41.3%) | |
| Consolidation chemotherapy | 22 (27.8%) | 25 (33.8%) | 15 (23.8%) | |
| Platform stage | 14 (17.7%) | 7 (9.4%) | 6 (9.5%) | |
| Unclear | 2 (2.5%) | 4 (5.4%) | 1 (1.6%) | |
| Recurrence | 1 (1.3%) | 5 (6.8%) | 10 (15.9%) | |
| Use of painkillers, antidepressants or hypnotic drugs | | | | <0.001 |
| No | 71 (89.9%) | 59 (79.7%) | 39 (61.9%) | |
| Yes | 8 (10.1%) | 15 (20.3%) | 24 (38.1%) | |
| Complications | | | | |
| No | 2 (2.5%) | 1 (1.4%) | 2 (3.2%) | 0.860 |
| Multiple myeloma bone disease | 20 (25.3%) | 41 (55.4%) | 36 (57.1%) | <0.001 |
| Impairment of renal function | 9 (11.4%) | 9 (12.2%) | 14 (22.2%) | 0.143 |
| Anaemia | 17 (21.5%) | 26 (35.1%) | 28 (44.4%) | 0.014 |
| Hypercalcaemia | 2 (2.5%) | 0 (0.0%) | 3 (4.8%) | 0.197 |
| Thrombosis | 1 (1.3%) | 3 (4.1%) | 1 (1.6%) | 0.531 |
| Inflammation | 16 (20.3%) | 16 (21.6%) | 8 (12.7%) | 0.355 |
| Other complications | 28 (35.4%) | 13 (17.6%) | 12 (19.0%) | 0.017 |
| Treatments | | | | 0.804 |
| Oral drug therapy | 3 (3.8%) | 4 (5.4%) | 1 (1.6%) | |
| Chemotherapy | 64 (81.0%) | 55 (74.3%) | 51 (81.0%) | |

Continued

**Table 4** Continued

| Variable | Class 1 (n=79) | Class 2 (n=74) | Class 3 (n=63) | P value |
|---|---|---|---|---|
| Autologous haematopoietic stem cell transplantation | 5 (6.3%) | 9 (12.2%) | 5 (7.9%) | |
| Others | 7 (8.9%) | 6 (8.1%) | 6 (9.5%) | |
| BMI | | | | 0.198 |
| <18.5 | 6 (7.6%) | 2 (2.7%) | 6 (9.5%) | |
| 18.5–23.9 | 50 (63.3%) | 52 (70.3%) | 34 (54.0%) | |
| 24–27.9 | 20 (25.3%) | 13 (17.6%) | 19 (30.2%) | |
| >27.9 | 3 (3.8%) | 7 (9.5%) | 4 (6.3%) | |
| Daily sleep time (mean±SD), hours | 7.2±1.4 | 6.7±1.4 | 6.5±1.5 | 0.014 |
| Daily step counts | | | | 0.043 |
| <5000 | 49 (62.0%) | 54 (73.0%) | 51 (81.0%) | |
| ≥5000 | 30 (38.0%) | 20 (27.0%) | 12 (19.0%) | |

BMI, body mass index; MM, multiple myeloma.

## DISCUSSION

To the best of our knowledge, this is the first study to describe the homogeneous symptoms and associated factors in patients with MM using routinely captured PROs. Our findings revealed that a high prevalence of burdensome symptoms was found in this population during treatment, and 63.4% of patients reported different degrees of symptom burden, which deserves clinical attention.

The current study identified the target population with the most severe symptoms. Classes 2 and 3 represented groups with greater symptom burden, with patients in class 3 being at a higher risk of symptom burden than those in class 2. Depression, fatigue and pain were identified as the most common symptoms in this study, consistent with the broader literature.[14 15] Boland *et al*[14] reported that fatigue and bone pain were the most common symptoms in patients undergoing treatment

**Table 5** Results of multinomial logistic regression of latent class membership

| Characteristics | Class 2 (n=74) | | | Class 3 (n=63) | | |
|---|---|---|---|---|---|---|
| | OR | 95% CI | P value | OR | 95% CI | P value |
| Monthly family income | | | | | | |
| ≥5000 | Ref | | | Ref | | |
| <5000 | 3.14 | 1.32 to 7.44 | 0.010 | 2.33 | 0.95 to 5.70 | 0.064 |
| Use of painkillers, antidepressants or hypnotic drugs | | | | | | |
| No | Ref | | | Ref | | |
| Yes | 1.66 | 0.59 to 4.68 | 0.338 | 3.68 | 1.34 to 10.11 | 0.012 |
| Complication (multiple myeloma bone disease) | | | | | | |
| No | Ref | | | Ref | | |
| Yes | 2.95 | 1.11 to 7.83 | 0.029 | 2.81 | 0.98 to 8.07 | 0.055 |
| Anaemia | | | | | | |
| No | Ref | | | Ref | | |
| Yes | 1.52 | 0.58 to 4.00 | 0.398 | 2.10 | 0.75 to 5.86 | 0.158 |
| Other complications | | | | | | |
| No | Ref | | | Ref | | |
| Yes | 0.75 | 0.28 to 2.00 | 0.563 | 1.04 | 10.36 to 3.00 | 0.946 |
| Daily sleep time | 0.83 | 0.64 to 1.07 | 0.143 | 0.74 | 0.57 to 0.97 | 0.031 |
| Daily step counts | | | | | | |
| ≥5000 | Ref | | | Ref | | |
| <5000 | 1.59 | 0.74 to 3.42 | 0.234 | 2.52 | 1.05 to 6.05 | 0.039 |

**Table 6** Functional status across the identified latent classes (mean±SD)

| Domain | Class 1 (n=79) | Class 2 (n=74) | Class 3 (n=63) | P value |
|---|---|---|---|---|
| Physical function | 50.2±6.7 | 44.0±4.1 | 37.8±7.9 | <0.001 |
| Ability to participate in social roles and activities | 56.0±8.0 | 49.0±4.1 | 41.6±6.0 | <0.001 |
| Cognitive function | 50.5±8.6 | 44.8±5.5 | 39.0±5.2 | <0.001 |

for MM. A study on patients with stable but advanced-stage MM also confirmed that fatigue and pain were the most frequent symptoms.[15] Furthermore, Naegele et al[16] reported that fatigue, diarrhoea and decreased appetite were the most frequent and distressing physical symptoms in patients with MM treated with autologous stem cell transplantation, and these symptoms differed substantially before and after treatment. This discrepancy may be caused by differences in the treatment of patients. A systematic review identified pain, fatigue, sleep issues and depression as the most frequently investigated symptoms in patients with MM.[17] Therefore, this study examined the association between demographic and cancer-related factors associated with different symptom profiles in patients with MM.

The demographic and cancer-related factors associated with different symptom profiles were identified in this study. An increase in age was not associated with a significant between-group difference. The median age in this sample was 60.3 years, which was relatively younger than the average age of patients with MM (73 years). Therefore, future studies should validate the results of this study in older patients. Complications of MM bone disease were predictive factors for high anxiety, depression and pain interference. These findings are consistent with those of a previous study by Ebraheem et al,[18] suggesting that increased comorbidities and myeloma-related bone and kidney damage were associated with a higher rate of moderate to severe symptoms. Furthermore, depressive symptoms were found to increase the incidence of a higher number of comorbidities in patients with MM aged ≥65 years.[19]

Consistent with previous reports, this study found that poor socioeconomic status was associated with increased symptoms of pain, suggesting that these patients may require additional support.[20] Low monthly family income was correlated with severe symptoms, particularly in the anxiety, depression and pain domains. Lower-income and unemployment have been reported as risk factors for financial burden as a direct result of cancer and related treatment and can be referred to as financial toxicity.[20] Previous studies have demonstrated that patients with MM have substantial areas of concern about the financial burden of disease (44%) and inability to perform daily activities (33%) and felt unhappy (41%). In addition, 36% reported pain and 15% reported depression,[21] highlighting the need for multidisciplinary psychosocial support. According to Ramsenthaler et al, several patients with MM were burdened by financial worries, almost 50% of patients with MM reported concerns about dying, and almost 80% had financial difficulties, of which 43.3% were severe.[17] Psychosocial stressors, including depression, cause high resistance to cell death, compromised immunity, and inflammatory and oxidative stress, which may affect cancer progression.[22] Additionally, these symptoms are associated with longer hospitalisation, low medication adherence and suicide.[22] Therefore, patients with poor socioeconomic status deserve more attention, particularly those with psychosocial symptoms. The use of painkillers, antidepressants or hypnotic drugs and low physical activity were significant predictors of a high symptom burden in patients with MM. Patients using these drugs are more likely to experience severe pain, depression and sleep issues and require medical attention than those who are not using these drugs.

Consistent with the results of previous studies, patients with greater symptom burden were more likely to report functional limitation than patients without burdensome symptoms.[15 17 23] For example, symptoms such as fatigue and pain have a negative effect on physical function and may cause treatment discontinuation.[15 23] In our study, patient physical function was slightly impaired compared with that of the general population, and significant differences were identified among the three classes. These results were consistent with those reported by Ramsenthaler et al,[17] suggesting that severely decreased social, physical and cognitive functions were common among patients with MM. However, the class with a lesser symptom burden was more likely to report high function levels, which may be due to selection bias. Differences in age may have contributed to these disparities, since relatively younger patients were included in this study. According to a national report in China, the mean age of patients diagnosed with MM was 58 years, which was approximately 10 years younger than that of Caucasians.[24] Since relatively young patients were included in this study, future studies should validate whether the findings can be generalised to older MM patient populations. Age is a well-established risk factor for cognitive decline, and impaired cognitive function is common in older patients. Older patients have been more vulnerable to cognitive side effects of cancer treatments than younger patients.[25] Support for an interaction of age, cognitive reserve and exposure to chemotherapy have been identified as risk factors for cognitive decline.[26] Hsu et al reported that mild anxiety may be associated with better cognitive performance, and higher anxiety may be associated with poorer cognitive performance.[27] In a longitudinal prospective

study of older patients with breast cancer, half of the investigated patients perceived a decline in cognitive function after 6 months of chemotherapy. However, there was no significant change in Mini-Mental Status Examination scores after chemotherapy or hormonal therapy over a short time in Janelsins *et al*'s study.[28] There is a lack of study regarding the effects of cancer treatment on a cognitive performance in older patients with MM. Therefore, prospective, long-term studies are necessary to assess the effects of MM treatment on the cognitive function of older patients.

## Limitations of the study

This study has some limitations. First, this was a cross-sectional study that did not explore the evolution of symptoms over time. Second, this study involved only two single centres, and the results need to be validated in multicentre studies. Moreover, the severity of the symptoms may have been underestimated, as some patients may have been significantly ill to participate in this study. Although the detail of anti-myeloma therapeutics could play an important role in the symptom experiences of MM patients, we did not specify the detail of chemotherapy. Finally, this study did not evaluate treatment response. Further studies are required to incorporate more diverse sampling to fill ongoing knowledge gaps in this field.

## Future studies

In future studies, a longitudinal study design should be considered first to present the dynamic changes in patient symptom characteristics over time and over the course of disease. Second, multiple data collection methods (such as paper, internet or telephone collection) should be implemented to protect the integrity of the data. It is necessary to implement symptom management based on PROs in heterogeneous populations.

## CONCLUSION

Three distinct symptom profiles were observed in patients with MM during treatment. Patients with low monthly family income and complications of MM bone disease were more likely to report high levels of anxiety, depression and pain interference than patients with high monthly family income and without complications of MM bone disease. Taking painkillers, antidepressants or hypnotic drugs, with daily step counts <5000, and with functional limitations increased the risk of developing high symptom burden. The identification of patients with high symptom burden management needs should focus on assessing demographic and clinical characteristics and functional status so that interventions can be designed for high-risk individuals.

**Author affiliations**
[1]Department of Hematopathology, The Second Affiliated Hospital of Guilin Medical University, Guilin, Guangxi, China
[2]Fudan University School of Nursing, Shanghai, China
[3]Department of Nephrology, Affiliated Hospital of Guilin Medical University, Guilin, Guangxi, China
[4]Guangxi Normal University, Guilin, Guangxi, China
[5]Department of Nursing, Peking Union Medical College Hospital, Beijing, China

**Acknowledgements** We thank the clinical staff at the study sites who enabled us to recruit the patients.

**Contributors** CYuan, as guarantor, had full access to all of the data in the study and takes responsibility of the data and the accuracy of the data analysis, and critical revision of the manuscript for important intellectual content. CYu, CYuan and TC designed the study. CYu, NZ, GP and WO were the principal investigators. TC and CYu wrote the manuscript. CYu, TC, XL and TZ performed the statistical analyses, and all authors reviewed and contributed to the manuscript. All authors critically revised the manuscript and provided final approval.

**Funding** This work was supported by the Science and Technology Project of Guilin City (20210227-8-5) and Project of the Guangxi Education Department (2021KY0521).

**Competing interests** None declared.

**Patient and public involvement** Patients and/or the public were not involved in the design, or conduct, or reporting, or dissemination plans of this research.

**Patient consent for publication** Not applicable.

**Ethics approval** This study involves human participants and ethical approval was obtained from the Institutional Review Board of the Second Affiliated Hospital of Guilin Medical University and all study sites (No. GLKJJH010). Participants gave informed consent to participate in the study before taking part.

**Provenance and peer review** Not commissioned; externally peer reviewed.

**Data availability statement** Data are available upon reasonable request. All data relevant to the study are included in the article or uploaded as supplementary information. The data that support the findings of this study are available upon reasonable request from the corresponding author.

**ORCID iD**
Changrong Yuan http://orcid.org/0000-0001-8480-2569

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
