## [Reviewer comments · BMJ Open]

This paper was submitted to a another journal from BMJ but declined for publication following peer review. The authors addressed the reviewers' comments and submitted the revised paper to BMJ Open. The paper was subsequently accepted for publication at BMJ Open.

ARTICLE DETAILS

TITLE (PROVISIONAL)	Classification of symptom subtypes in patients with multiple myeloma during treatment: A cross-sectional survey study in China
AUTHORS	Yu, Chunfang; Cai, Tingting; Zhou, Tingting; Zeng, Ning; Liang, Xin; Pan, Guihua; Ouyang, Wei; Yuan, Changrong

VERSION 1 – REVIEW

REVIEWER	Suzuki, Kazuhito Jikei University School of Medicine, Clinical Oncology and Hematology
REVIEW RETURNED	25-Aug-2022

GENERAL COMMENTS	Subtypes of symptoms in patients with multiple myeloma during treatment: A latent class analysis approach. The article " Subtypes of symptoms in patients with multiple myeloma during treatment: A latent class analysis approach" (bmjopen-2022-066467) by Yu C, et al. investigated that several symptoms were associated with patients' backgrounds, disease status, and treatment using PROMIS-57 prospectively. This study design was very interesting, but there are several issues to be addressed as below. Major issues 1. Patient characteristics were not enough. For instance, "disease staging" show during treatment or not, and recurrence or not. I considered that therapeutics and treatment response should affect symptom. Therefore, the author should describe patient characteristics more in detail classified "disease status (ISS stage)" "treatment (PI, IMiDs, monoclonal antibodies, steroids, cytotoxic agents, and transplantation)" "treatment response (CR, PR, SD, and PD)". In addition, how to decide the cutoff values of monthly family income and daily step counts was also unclear. The author could add several comments about it.2. Mean cognitive function score was significant different among 3 groups in table 5. However, "No significant decrease was identified in social and cognitive aspects." In discussion. Which was correct? If there was a significant difference of cognitive function scores among these 3 groups, I considered that it was very interesting from point of pathogenesis of cognitive disorder in myeloma
--

	patients. I highly recommended that the author should discuss more the association between cognitive disorder and clinical symptom in myeloma patients. Minor issues  1. "Daily sleep time" was mentioned in table 3, but not shown in table 2, and 4. 2. How to define "subgroup description" was unclear. The author should mention about that in methods.
--	--

REVIEWER	Petruski-Ivleva, Natalia Aetion, Inc., Science
REVIEW RETURNED	26-Aug-2022

GENERAL COMMENTS	I am not sure study findings present actionable findings. I would have preferred to see descriptive statistics and the distribution of patients reporting each evaluated symptom, as well as the distribution of scores for all items assessed by the PROMISE tools. The potential predictors presented in table 4 are not very informative, with exception of income. It is well known that patients with bone disease experience pain. The use of painkillers and antidepressants should not be used to predict pain and anxiety, as the use of medications is a consequence of those symptoms. Daily step counts are a consequence and not a predictor of more symptomatic disease presentation. I would prefer to see associations with individual demographics, clinical complications, which as anemia and hypercalcemia, as well as treatment categories. In table 5, it appears that patients with fewer symptoms had higher scores for functional status indicating worse performance. These results are not intuitive. I would recommend revising the content of the paper to present descriptive results only. Presented associations do not add to current understanding or management of the disease.
--

VERSION 1 – AUTHOR RESPONSE

1. Please revise the title of your manuscript to include the research question, study design and setting. This is the preferred format of the journal.

Response: Thank you for this comment. We have revised the title of the manuscript as follows:
Classification of symptom subtypes in patients with multiple myeloma during treatment: A cross-sectional survey study in China

2. Please revise the 'Strengths and limitations of this study' section of your manuscript (after the abstract). This section should contain up to five short bullet points, no longer than one sentence each, that relate specifically to the methods. The novelty, aims, results or expected impact of the study should not be summarized here.

Response: Thank you for this comment. We have revised the section on 'Strengths and limitations of this study (page 2) as follows:

Strengths and limitations of this study

A latent class analysis was used to categorize cancer-related symptoms in Chinese patients undergoing treatment for multiple myeloma.

This study assessed between-group differences in the demographic and clinical characteristics and functional status of patients with multiple myeloma.

- The Patient-Reported Outcomes Information System (PROMIS)-57 and the PROMIS Cognitive Function Short Form were used as research tools.
- This study involved a limited number of treatment centers.
- Patient clinical characteristics did not include the evaluation of treatment response.

3. Please complete a thorough proofread of the text and correct any spelling and grammar errors that you identify.

Response: We have completed a thorough proofread of the text and revised the manuscript accordingly.

4. In the methods you state “a convenience sample of patients with MM in the Department of Hematology of a tertiary hospital in Guilin, China.” However, elsewhere in the manuscript you state that the study took place in two hospitals. Please ensure these statements are consistent throughout the manuscript.

Response: Thank you for this comment. We have ensured consistency in the description of the methods in the revised manuscript and confirm that the study took place in two tertiary hospitals in Guilin, China. We have revised the manuscript as follows (Methods section, page 4, paragraph 3): This study adopted a cross-sectional study design and adhered to the Strengthening The Reporting of OBservational Studies in Epidemiology guidelines. Convenience sampling was used to recruit patients with MM visiting the hematology department of two tertiary hospitals affiliated with Guilin Medical University in China between July and December 2021.

5. Please ensure that the Abstract in ScholarOne and in the main document should be the same.

Response: Thank you for your advice. We have ensured that the Abstract in ScholarOne is the same as that in the main document.

Reviewer reports:

Reviewer(s)' Comments to Author:

Reviewer: 1

Comments to the Author:

Major issues

1. Patient characteristics were not enough. For instance, “disease staging” show during treatment or not, and recurrence or not. I considered that therapeutics and treatment response should affect symptom. Therefore, the author should describe patient characteristics more in detail classified “disease status (ISS stage)” “treatment (PI, IMiDs, monoclonal antibodies, steroids, cytotoxic agents, and transplantation)” “treatment response (CR, PR, SD, and PD)”. In addition, how to decide the cutoff values of monthly family income and daily step counts was also unclear. The author could add several comments about it.

Response: Thank you for the comments. Although we did consider multiple clinical variables in the manuscript, we did not collect data on treatment response. This is a limitation of the study design and is mentioned in the Limitations section (page 2, paragraph 1). We will take this suggestion into account in our future longitudinal study design. However, we did make some revisions in the patient's characteristics to present more clinical variables that we have collected (Results section, page 7, paragraph 1, Patient characteristics): In terms of clinical characteristics, most patients had IgG MM (55.1%), were undergoing chemotherapy (70.8%), and had multiple concurrent complications of treatment (97.7%). Regarding treatment options, most patients underwent chemotherapy (78.7%) and did not require analgesia, antidepressants, or hypnotics before the diagnosis (97.7%). Most patients had a BMI within the normal range (63.0%) and reported daily step counts <5000 steps (71.3%). Family monthly income is defined according to the level of local per capita income, which is a commonly used domestic division standard. The daily step counts were defined by reference to the moderate intensity of aerobic exercise commonly used in China.

2. Mean cognitive function score was significant different among 3 groups in table 5. However, “No significant decrease was identified in social and cognitive aspects.” In discussion. Which was correct? If there was a significant difference of cognitive function scores among these 3 groups, I considered that it was very interesting from point of pathogenesis of cognitive disorder in myeloma patients. I highly recommended that the author should discuss more the association between cognitive disorder and clinical symptom in myeloma patients.

Response: Thank you for your valuable comments. We have checked the data in Table 6 and the statements in the Discussion section (Results section, page 15-16, paragraph 2). The T-scores for physical function, ability to participate in social roles and activities, and cognitive function were relatively low in the sample according to the score manual of PROMIS-57, in which physical function was slightly impaired when compared with the general population. Significant differences were identified among the three classes, in which Class 1 scored the highest in all functional dimensions, whereas Class 3 scored the lowest ($P < 0.05$).

Age is a well-established risk factor for cognitive decline, and impaired cognitive function is common in older patients. Older patients have been more vulnerable to cognitive side effects of cancer treatments than younger patients [25]. Support for an interaction of age, cognitive reserve, and exposure to chemotherapy have been identified as risk factors for cognitive decline [26]. Hsu et al. reported that mild anxiety may be associated with better cognitive performance, and higher anxiety may be associated with poorer cognitive performance [27]. In a longitudinal prospective study of older patients with breast cancer, half of the investigated patients perceived a decline in cognitive function after 6 months of chemotherapy. However, there was no significant change in Mini-Mental Status Examination scores after chemotherapy or hormonal therapy over a short time in Janelins' study [28]. There is a lack of study regarding the effects of cancer treatment on a cognitive performance in older patients with MM. Therefore, prospective, long-term studies are necessary to assess the effects of MM treatment on the cognitive function of older patients.

Minor issues

1. “Daily sleep time” was mentioned in table 3, but not shown in table 2, and 4.

Response: Thank you for raising this point. We have added Table 3 to the revised manuscript and Table 3 in the original manuscript has been renamed Table 4. Tables 2 and 4 in the original manuscript were renamed to the revised Tables 3 and 5, respectively. Table 2 mainly concerned the model fit statistics for each of the fitted latent class analysis models. Daily sleep time was mentioned in Table 5 because the “Daily sleep time” was treated as a continuous variable in this study.

2. How to define “subgroup description” was unclear. The author should mention about that in methods.

Response: Thank you for raising this point. The subgroup description has been clarified and is described in the manuscript (Results section, page 9, paragraph 1): The item probabilities of the latent classes are shown in Figure 1. Class 1 was characterized by low item probabilities for all symptom aspects and was labeled as the “low symptom burden group” ($n=79$, 36.6%). Class 2 was characterized by elevated item probabilities for anxiety, depression, and pain interference, whereas decreased item probabilities were associated with fatigue and sleep disturbance. Therefore, Class 2 was labelled as the “moderate symptom burden group” ($n=74$, 34.2%). Class 3 was characterized by high item probabilities for all symptoms and was labeled as the “high symptom burden group” ($n=63$, 29.2%).

Reviewer: 2

Comments to the Author :

1. I am not sure study findings present actionable findings. I would have preferred to see descriptive statistics and the distribution of patients reporting each evaluated symptom, as well as the distribution of scores for all items assessed by the PROMIS tools.

Response: Thank you for raising these issues. The findings of this study will help to identify the high risk population of patients and give precise intervention. Patients with MM experienced varying degrees of symptoms during treatment. The identification of patients with high symptom burden management needs should focus on assessing demographic and clinical characteristics and functional status so that interventions can be designed for high-risk individuals. We have added the distribution of scores for symptoms assessed by the PROMIS tool in the revised manuscript (Table 3):

(Results section, page 9, paragraph 2):

Between-group differences in symptom status across the latent classes

A comparison of symptoms across the latent classes is reported in Table 3. The T-scores for anxiety, depression, fatigue, sleep disturbance, and pain interference were relatively lower in the sample according to the score manual of PROMIS-57 compared with the general population. There were significant differences in the mean T-scores for each domain among the latent classes, with the highest scores observed in Class 3 and the lowest scores observed in Class 1 ($P < 0.05$).

2. The potential predictors presented in table 4 are not very informative, with exception of income. It is well known that patients with bone disease experience pain. The use of painkillers and antidepressants should not be used to predict pain and anxiety, as the use of medications is a consequence of those symptoms. Daily step counts are a consequence and not a predictor of more symptomatic disease presentation. I would prefer to see associations with individual demographics, clinical complications, which as anemia and hypercalcemia, as well as treatment categories.

Response: Thank you for your suggestions. We have added Table 3 to the revised manuscript and Table 4 in the original manuscript has been renamed Table 5. This study aimed to classify subgroups of cancer-related symptoms in patients with multiple myeloma (MM) during treatment and examine between-group differences in demographic and clinical characteristics in addition to functional status. We did not include individual patient demographics or clinical complications and these limitations have been acknowledged in the limitations section in the main manuscript. We have added more description in terms of the associations with individual demographics, clinical complications, and treatment categories.

3. In table 5, it appears that patients with fewer symptoms had higher scores for functional status indicating worse performance. These results are not intuitive.

Response: Thank you for these comments. We have added Table 3 to the revised manuscript and Table 5 in the original manuscript has been renamed Table 6. The T-scores for physical function, ability to participate in social roles and activities, and cognitive function were relatively low in the sample according to the score manual of PROMIS-57, in which physical function was slightly impaired when compared with the general population. There were significant differences in the mean T-scores for each domain among the latent classes, with the highest scores observed in Class 3 and the lowest scores observed in Class 1 ($P < 0.05$).

We have added Figure 2 in order to present the results in a more intuitive way. An explanation for the results has been added (Discussion section, page 15, paragraph 2): Consistent with the results of previous studies, patients with greater symptom burden were more likely to report functional limitation than patients without burdensome symptoms [15,17,23]. For example, symptoms such as fatigue and pain have a negative effect on physical function and may cause treatment discontinuation [15,23]. In our study, patient physical function was slightly impaired compared with that of the general population, and significant differences were identified among the three classes. These results were consistent with those reported by Ramsenthaler et al. [17], suggesting that severely decreased social, physical, and cognitive functions were common among patients with MM. However, the class with lesser symptom burden was more likely to report high function levels, which may be due to selection

bias. Differences in age may have contributed to these disparities, since relatively younger patients were included in this study. According to a national report in China, the mean age of patients diagnosed with MM was 58 years, which was approximately 10 years younger than that of Caucasians [24]. Since relatively young patients were included in this study, future studies should validate whether the findings can be generalized to older MM patient populations.

4. I would recommend revising the content of the paper to present descriptive results only.

Response: Thank you for your suggestion. Descriptive results have been added accordingly. For example, we have added Table 3 and Figure 2 in the revised manuscript.

VERSION 2 – REVIEW

REVIEWER	Suzuki, Kazuhito Jikei University School of Medicine, Clinical Oncology and Hematology
REVIEW RETURNED	28-Nov-2022

GENERAL COMMENTS	The article " Classification of symptom subtypes in patients with multiple myeloma during treatment: A cross-sectional survey study in China" (bmjopen-2022-066467.R1) was revised well according to reviewers' comments. However, the most important issue was still unclear. The detail of anti-myeloma therapeutics might play the most important role to investigate several symptoms of myeloma patients. Therefore, the author should add the detail of chemotherapies, such as proteasome inhibitors, immunomodulatory drugs, steroids, or cytotoxic agents containing regimens, if possible.
--

VERSION 2 – AUTHOR RESPONSE

We are grateful to the reviewers and the editor for their time in reviewing our manuscript and providing feedback. We have carefully considered the comments and revised the manuscript accordingly. Please find below our point-by-point responses to the editor's and reviewers' comments. For your convenience, the comments are set out in black font while our responses are set out in blue font.

Reviewer: 1

Comments to the Author:

The article " Classification of symptom subtypes in patients with multiple myeloma during treatment: A cross-sectional survey study in China" (bmjopen-2022-066467.R1) was revised well according to reviewers' comments. However, the most important issue was still unclear. The detail of anti-myeloma therapeutics might play the most important role to investigate several symptoms of myeloma patients. Therefore, the author should add the detail of chemotherapies, such as proteasome inhibitors, immunomodulatory drugs, steroids, or cytotoxic agents containing regimens, if possible.

Response: Thank you for the comments. Although we did consider multiple clinical variables in the manuscript, we did not collect data on the detail of chemotherapy. This is a limitation of the study design and is mentioned in the Limitations section. We will take this suggestion into account in our future longitudinal study design. We have revised the manuscript as follows (Limitations section, page 16, paragraph 2): Although the detail of anti-myeloma therapeutics could play an important role in the symptom experiences of MM patients, we did not specify the detail of chemotherapy.